# Cross-lingual QA:
# A Key to Unlocking In-context Cross-lingual Performance

**Sunkyoung Kim** [* 1]   **Dayeon Ki** [* 2]   **Yireun Kim** [1]   **Jinsik Lee** [1]

## Abstract

Multilingual large language models (MLLMs) have demonstrated significant cross-lingual capabilities through in-context learning. Existing approaches typically construct monolingual in-context examples, either in the source or target language. However, translating entire in-context examples into the target language might compromise contextual integrity and be costly in the case of long-context passages. To address this, we introduce Cross-lingual QA, a cross-lingual prompting method that translates only the question and answer parts, thus reducing translation costs. Experiments on four typologically diverse multilingual benchmarks show that Cross-lingual QA prompting effectively stimulates models to elicit their cross-lingual knowledge, outperforming prior monolingual prompting approaches. Furthermore, we show that prompting open-source MLLMs with cross-lingual in-context examples enhances performance as the model scale increases.

## 1. Introduction

In-context learning capabilities of Large Language Models (LLMs) have been extensively studied since the successful emergence of GPT-3 (Brown et al., 2020). This success extends to multilingual LLMs (MLLMs) such as BLOOM (Scao et al., 2022) and XGLM (Lin et al., 2022), which aim to enhance in-context learning performance across various multilingual tasks. However, not all languages are adequately addressed during the pre-training stage, leading to inconsistent cross-lingual task performance across different languages.

Previous approaches typically use translation to enhance

task performance in the target language (Conneau et al., 2020; Liu et al., 2019; Xue et al., 2021). One of the earliest methods, *translate-train*, involves translating the train dataset into the target language. More recently, LLMs have utilized translated in-context examples (Ahuja et al., 2023; Asai et al., 2023) to evaluate cross-lingual task performance. However, translating entire task datasets can interfere with the integrity of the context and incur high translation costs, especially for lengthy passages.

To address these issues, we propose **Cross-lingual QA** prompting, which maintains the passage in the source language while translating only the question and answer pairs into the target language, as illustrated in Figure 1. We find that selectively translating certain components of the dataset is an efficient method for enhancing cross-lingual task performance. Retaining the passage in the source language and translating only the question-answer pairs in the in-context examples achieves performance comparable to fully translated counterparts.

We experiment with four multilingual benchmarks across three task categories: classification, reasoning, and question answering (QA). We evaluate XNLI (Conneau et al., 2018) for classification and XCOPA (Ponti et al., 2020) for reasoning. For QA tasks, we use XQuAD (Artetxe et al., 2020) and MLQA (Lewis et al., 2020), which test the model's comprehension ability to answer questions using given contexts. Experimental results on these tasks demonstrate the effectiveness of the Cross-lingual QA prompt, matching the performance of entirely translated in-context examples. Additionally, we observe that the effect of our Cross-lingual QA prompt improves significantly with larger model sizes.

To summarize, our contributions are as follows:

- We introduce **Cross-lingual QA** prompting, which constructs in-context examples by retaining the passage in the source language and translating the question-answer pairs into the target language.

- Cross-lingual QA prompting demonstrates remarkable in-context cross-lingual performance with MLLMs, requiring minimal translation compared to existing prompting methods.

---
*Equal contribution. Work done during internship at LG AI Research. [1]LG AI Research [2]University of Maryland. Correspondence to: Jinsik Lee <jinsik.lee@lgresearch.ai>.

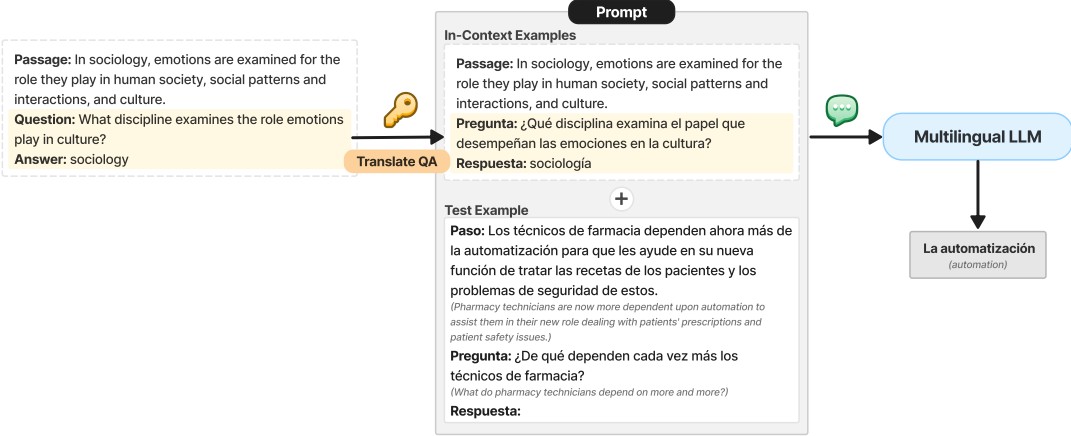

*Figure 1.* Overview of the **Cross-lingual QA** prompting method. In each in-context example, the passage remains in the source language while the question and answer are translated into the target language. **Gray** text in brackets represents English translations. The test example is always presented in the target language. We concatenate $k$ in-context examples and the test example to form an in-context example, which is then fed into a multilingual LLM.

## 2. Related Work

**In-context Learning.** Brown et al. (2020) first propose in-context learning (ICL) as a branch of meta-training[1]. In-context examples used to measure ICL are typically constructed by concatenating instructions with a test example, which is then fed into the model for generation (Radford et al., 2019). ICL is widely recognized as an effective approach for evaluating large-scale models as it does not require any parameter updates.

**Cross-lingual Performance.** An extension of ICL in a cross-lingual setting aims to measure the transferability of MLLMs from the source language to the target language. In this context, in-context examples are composed in the source language while the test example is in the target language. Previous work has explored ICL with cross-lingual transfer (Winata et al., 2021; Ahuja et al., 2023; Asai et al., 2023) and multilingual Chain of Thoughts (CoT) prompting (Shi et al., 2023; Huang et al., 2023). However, these works mainly compose the in-context examples in a single language. Further, translating entire in-context examples into the target language is expensive, especially for lengthy passages. Our research aligns with efforts to improve cross-lingual performance of MLLMs; we mix both source and target languages in the in-context examples to elicit better cross-lingual alignment.

[1]In the context of LLMs, meta-training means that models first develop a range of skills during training and utilize those abilities during inference for a target task (Brown et al., 2020).

## 3. Cross-lingual QA Prompting

Existing approaches often rely on translating entire data instances, such as translating the training dataset (*translate-train*) or the test dataset (*translate-test*) (Hu et al., 2020). Ahuja et al. (2023) show that *translate-test* achieves higher performance compared to using target language examples. However, this approach depends heavily on the English performance of the tested model, making it difficult to measure its cross-lingual capability precisely.

| **Model** | $(Q_{\text{tgt}}, A_{\text{tgt}})$ | $(Q_{\text{tgt}}, A_{\text{src}})$ | $(Q_{\text{src}}, A_{\text{tgt}})$ |
|---|---|---|---|
| XGLM | **41.53** | 40.07 | 37.37 |
| BLOOM | **37.06** | 36.42 | 35.49 |

*Table 1.* Comparison of accuracy scores on XQuAD test set for different prompt designs. $(Q_{\text{tgt}}, A_{\text{src}})$ represents our Cross-lingual QA prompt.

**Prompt Design.** To design the Cross-lingual QA prompt, we first compare prompts with different question and answer language combinations using the XQuAD dataset. We fix the language of the passage in the source language. Then, we compare three variants: **(1)** Question in source language and Answer in target language $(Q_{\text{src}}, A_{\text{tgt}})$, **(2)** Question in target language and Answer in source language $(Q_{\text{tgt}}, A_{\text{src}})$, and **(3)** both Question and Answer in target language $(Q_{\text{tgt}}, A_{\text{tgt}})$.

As shown in Table 1, translating both the question and answer into the target language $(Q_{\text{tgt}}, A_{\text{tgt}})$ achieves the best

| Task | Dataset | # Validation set | # Test set | Languages | # Lang. | Metric |
|---|---|---|---|---|---|---|
| **Classification** | XNLI | 2,490 | 5,010 | ar, bg, de, en, el, es, fr, hi ru, sw, tr, th, ur, vi, zh | 15 | EM |
| **Reasoning** | XCOPA | 100 | 500 | en, et, ht, id, it, sw, ta, th, tr, vi, zh | 11 | EM |
| **QA** | MLQA | 504 - 1,148 | 4,517 - 11,590 | ar, de, en, es, hi, vi, zh | 7 | F1 |
| | XQuAD | 50 | 1,190 | ar, de, en, el, es, hi, ro, ru, th, tr, vi, zh | 12 | F1 |

*Table 2.* Detailed statistics of 4 multilingual benchmarks. # Validation set and # Test set denotes the number of dataset instances per supported language. All language codes follow the ISO 639-1 Code.

performance. However, the other variants also demonstrate cross-lingual ability to some extent. This suggests that even a pinch of translation of the question or answer can effectively enhance target language task performance. Therefore, the Cross-lingual QA prompt translates the question-answer pair into the target language while anchoring the passage in the source language as in Figure 1.

## 4. Experimental Setup

### 4.1. Dataset

We evaluate model performance on four multilingual tasks, categorized into three types: **(1)** Classification (XNLI (Conneau et al., 2018)), **(2)** Reasoning (XCOPA (Ponti et al., 2020)), and **(3)** Question Answering (MLQA (Lewis et al., 2020) and XQuAD (Artetxe et al., 2020)). We select multilingual tasks with parallel dataset instances that cover typologically diverse languages. We fix English as the source language (En-XX). For each task, we use the validation set for in-context examples and the test set for evaluation at inference time. Detailed statistics for each dataset are provided in Table 2.

XQuAD (Artetxe et al., 2020) dataset only provides a test split, unlike the other tasks, which provide both validation and test splits. Originally, XQuAD is a parallel multilingual extractive QA dataset, where English sentences from a subset of SQuAD v.1 (Rajpurkar et al., 2016) are human-translated into 11 target languages (ar, de, el, es, hi, ro, ru, th, tr, vi, zh). To simulate this, we construct a quality-filtered, machine-translated validation set of XQuAD by translating a subset of the SQuAD v.1 train dataset with Google Translate API[2]. Detailed data construction process is described in Appendix A.

We design prompt templates in a Question Answering (QA) format following existing templates (Muennighoff et al., 2023; Shi et al., 2023) used to evaluate multilingual tasks.

---
[2]https://translate.google.com/

Our templates consist of the passage in the source language and the question-answer pair in the target language for in-context examples. For non-QA tasks (XNLI, XCOPA), we adapt the prompt templates to the QA format within the in-context examples. Further details of each prompt template are described in Appendix B.

### 4.2. Baseline

**Models.** Our study encompasses billion-scale variants of two open-source MLLMs. We generate outputs using greedy decoding.

- **XGLM** (Lin et al., 2022): Pre-trained on the CC-100 (Common Crawl) corpus, which includes 30 different languages. We use 1.7B, 2.9B, and 7.5B variants.

- **BLOOM** (Scao et al., 2022): Pre-trained on the ROOTS (Laurençon et al., 2022) corpus comprising 46 languages and 13 programming languages. We focus on variants similar in size to XGLM (1B, 3B, and 7B).

**Prompting Methods.** We evaluate three different prompting methods, distinguished by the language composition of the in-context examples. Note that we use the same test example in the target language across all three methods.

- **Source Language Prompting:** In-context examples are in the source language (English) and the test example in the target language, called as zero-shot cross-lingual transfer.

- **Target Language Prompting:** Both the in-context examples and the test example are in the target language.

- **Cross-lingual QA Prompting:** Our proposed prompting method described in Section 3. We only translate the QA pairs in the in-context examples into the target language.

| Models | Prompting Method | Classification | Reasoning | Question Answering | |
|---|---|---|---|---|---|
| | | XNLI (EM) | XCOPA (EM) | MLQA (F1) | XQuAD (F1) |
| **XGLM** | Source Language | 29.45 | 39.44 | 36.49 | 35.19 |
| | Target Language | 32.67 | 46.27 | 37.80 | 38.08 |
| | Cross-lingual QA | **32.74** (+0.07) | **48.06** (+1.79) | **39.34** (+1.54) | **41.70** (+3.62) |
| **BLOOM** | Source Language | 24.41 | 35.67 | 44.74 | 34.56 |
| | Target Language | 31.77 | 42.91 | **46.45** | 36.41 |
| | Cross-lingual QA | **32.00** (+0.23) | **43.93** (+1.02) | 45.42 | **37.22** (+0.81) |

*Table 3.* Average 5-shot performance of 7B variants over three random runs. For all prompting methods, the in-context examples are composed differently, but every test example is identical in the target language. Scores in **bold** indicate the best performance among the three prompting methods for each task. Green numbers in brackets denote the performance gain compared to Target Language prompting.

## 5. Results

**Overall.** For non-QA tasks, we measure the exact match (EM) of the generated answers, and for QA tasks, we measure the F1 score. As shown in Table 3, the Cross-lingual QA prompt consistently outperforms Source Language prompting, indicating that using question-answer pairs in the target language within the in-context examples enhances the model's cross-lingual transferability. Surprisingly, most Cross-lingual QA prompting results even surpass those of Target Language prompting. These findings validate that our method is crucial in building alignment between the source and target languages through in-context learning.

**Effect of Translation Method.** Both tested QA datasets are translated from the same English source: SQuAD v.1 (Rajpurkar et al., 2016). The validation set used for constructing in-context examples is human-annotated for MLQA, whereas we construct a quality-filtered machine-translated set for XQuAD (§4.1). Despite the differences in translation method, the overall trend is similar; Cross-lingual QA prompting consistently outperforms both Source and Target Language prompting. This suggests that using a few machine-translated QA pairs can significantly enhance a model's cross-lingual transferability without the need for expensive human annotations or translating entire examples.

**Model Performance at Scale.** We compare the average F1 performance for the XQuAD task across model size variants. As shown in Figure 2, cross-lingual transferability increases with model size. We observe that Cross-lingual QA prompting becomes more effective at the 7B scale, with consistent increases on average from 1B to 7B, following the scaling law (Kaplan et al., 2020). Cross-lingual QA prompts show the following performance gains: +7.25 and +4.27 points for BLOOM and XGLM, respectively.

Additionally, we demonstrate that larger models are more effective at narrowing the performance gap between Target Language and Cross-lingual QA prompting. We attribute

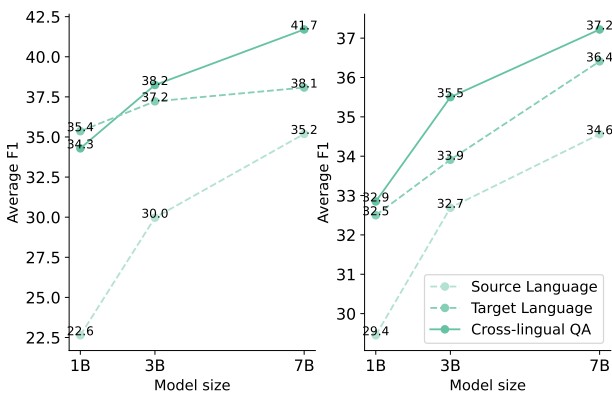

*Figure 2.* Average XQuAD performance for each prompting method at model scale. Model size on the $x$-axis is the approximated values for each model size variant. Left: XGLM, Right: BLOOM.

to Target Language prompting being easier to achieve with larger models, as it only requires understanding the target language, whereas Cross-lingual QA prompting assesses the model's understanding of both the source and target languages.

## 6. Conclusion

In this work, we explore the cross-lingual capabilities of existing multilingual language models through in-context learning. We propose a novel cross-lingual prompting method, **Cross-lingual QA**, which composes in-context examples by keeping the passage in the source language while translating the question-answer pairs into the target language. Our experiments highlight that translating only the question and answer into the target language in the in-context examples effectively stimulates the models to elicit their knowledge more *cross-lingually*.

# 7. Limitations

Our research primarily focuses on the cross-lingual capabilities of multilingual large language models. Recently, there are line of works showing that English-centric large language models also demonstrate multilingual abilities to some extent (Anil et al., 2023; Lai et al., 2023; Brown et al., 2020; Ahuja et al., 2023). It would be meaningful to extend our experiments to include English-centric large language models and compare their performance with multilingual models trained on various languages.

# Acknowledgements

We thank Hyeongu Yun, Hyunjik Jo for helpful discussions and feedback on our paper.

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

# A. XQuAD Data Construction

In this section, we detail the data preparation process for constructing the validation split for XQuAD (Artetxe et al., 2020). We construct parallel in-context examples for 11 target languages by first randomly selecting English examples from the SQuAD v.1 train set, then translating them into each target language using the Google Translate API. We build a dataset pipeline to obtain a high-quality machine-translated dataset.

Our dataset pipeline consists of five steps:

1. **Random Selection:** Randomly select English instances from the SQuAD v.1 train set.

2. **Translation:** Translate the selected instances using a widely-used open-source machine translation (MT) system.

3. **Quality Estimation:** Extract quality estimation candidates, including passages and question-answer pairs.

4. **High-Quality Candidate Selection:** Apply a round-trip translation (RTT) strategy to select high-quality candidates and place them into a candidate pool.

5. **Filteration:** Filter out instances with duplicate questions or with answers that are not included in the context.

First, we translate data instances from the randomly sampled SQuAD v.1 train set into 11 target languages using Google Translate. To address data quality issues from using an automatic MT system, we adopt a quality estimation process using the round-trip translation (RTT) strategy. RTT, also known as bi-directional translation, involves first translating into the target language (*Trans1*) and then back into the source language (*Trans2*).

Next, we extract contexts and questions for quality control candidates. We calculate BLEU scores for each candidate between *Trans1* and *Trans2*, similar to the method used for constructing cross-lingual summarization datasets (Zhu et al., 2019). We then construct a candidate pool by discarding data instances that do not meet our threshold (BLEU (Papineni et al., 2002) score $\geq 50$). From the candidate pool, we apply three filtering rules to refine the parallel extractive QA dataset: **(1)** Ensure unique ID values for all language variations, **(2)** Verify that the answer is included in the passage, and **(3)** Remove examples with duplicate questions. Finally, we randomly select a number of in-context examples per target language during in-context learning. We will release the entire dataset.

# B. Prompt Templates

For in-context evaluation, we adopt QA-formatted templates as shown in Table 4, which are based on templates from previous approaches (Muennighoff et al., 2023; Shi et al., 2023).

| Task | Prompt Template |
|---|---|
| **XNLI** | `Premise:`{premise}
`Question:`{hypothesis} `True, False, or Neither?`
`Answer:` |
| **XCOPA** | `Premise:`{premise}
`Question:What was the question?`
`Question:What happened as a result?`
`Options:`
`-`{choice1}
`-`{choice2}
`Answer:` |
| **MLQA, XQuAD** | `Passage:`{context}
`Question:`{question}
`Answer:` |

*Table 4.* Prompt templates to construct in-context examples for each task. For Cross-lingual QA prompting, underlined component is consistently written in the source language while other components are written in the target language.

