# OpenReview forum: "Cross-lingual QA: A Key to Unlocking In-context Cross-lingual Performance"
_ICML.cc/2024/Workshop/ICL — ICML 2024 Workshop ICL Poster_

### Official Review · Reviewer_gE9d · 2024-06-07
**Empirical Evaluation of Mutilingual-LLM In-context Performance with Prompt Engineering**

**Rating:** 2
**Fit:** 3
**Confidence:** 3

**Workshop Review:**

**Paper Summary**:

This paper presents an **empirical** evaluation of the **in-context** multilingual capabilities of two **multilingual** large language models across three scales. It leverages four diverse multilingual benchmarks, with the question-answer (QA) parts translated alongside the original prompts to create few-shot evaluation test sets. The authors conducted comparable ablations on different combinations of source-target languages for prompting translation, and the results demonstrate that the proposed method, with only the QA translated, yields the best performance. However, there are some limitations as listed in the following section.

**Reason For Not Giving Higher Score:**

**Summary Of Weaknesses**:

- Limited clarity and impact: The word 'Enough' in the title should have a clearer scope, specifically for multilingual LLMs and especially the two used in the paper. The authors could easily extend the scope to emerging English-centric LLMs or focus on model training besides evaluation to increase impact.
- Translation issues: Translated data quality is a significant concern even with the proposed RTT, such as loss of nuance and accuracy. Some features of the original prompts in the source language could be missed after translation, like tones, etc. All these factors might affect the results.
- Lack of expression consistency: For example, in the abstract, it mentions 'outperforming the full example translation prompting,' but in the following sections, it uses 'achieves comparable performance to fully translated in-context examples.' Some other cases are related to few-shot, repetitive introduction of the source-target prompting in different sections, etc.
- Benchmarks selection: MMLU multilingual and MGSM are better options for evaluating LLM multilingual capabilities.

**Reason For Not Giving Lower Score:**

**Summary Of Strengths**:

- Good direction to explore: Multilingual prompt engineering is a challenging topic, especially within the context of English-centric LLM pretraining, fine-tuning, and evaluation. Overall, this empirical evaluation provides valuable data points for **evaluation** work of both industrial and academic projects.
- Alignment with workshop topic: By translating only the QA part in the test set, this work explores unlocking the in-context capabilities of MLLM, aligning with the requirements of empirical evaluation of the performance of ICL on interpretability, controllability, and safety considerations for ICL systems.
- Comprehensive sections: The authors present all the necessary sections, diagrams, and tables to introduce the usefulness of their work and also provide limitations.

---

### Official Review · Reviewer_oNB5 · 2024-06-08

**Rating:** 2
**Fit:** 3
**Confidence:** 2

**Workshop Review:**

This study investigates the efficient way to do multilingual tasks through in-context learning. It suggests that solely translating the question and answer parts is sufficient to have a strong cross-lingual capability for multilingual large language models. It tests on four multilingual benchmarks across three task categories.

Pros:

- The proposed cross-lingual QA prompting is significantly more efficient than translating entire task datasets.
- It comprehensively examines the performance of LLMs across varying sizes.
- It considers multiple prompting methods.

Cons:

- The translation quality may influence the performance, especially for long source language contexts. This work doesn’t conduct detailed ablation study on the impact of translation quality.

Comments:

- Probably the authors can add some tasks, where the translation quality has minimal influence on the performance, such as math questions.

**Reason For Not Giving Higher Score:**

It lacks further analysis or ablation study on cross-lingual transferability.

**Reason For Not Giving Lower Score:**

The finding is meaningful, significantly reducing the cost of doing multilingual tasks.

---

### Meta-Review · Area_Chair_JGtd · 2024-06-14

**Recommendation:** 2

**Metareview:**

This paper explores the cross-lingual capabilities of existing multilingual language models through in-context learning and proposes a few-shot prompting method called Cross-lingual QA. The method suggest that translating only the question and answer parts is sufficient to have a strong cross-lingual capability in multilingual LLMs.

All reviewers have a favorable opinion of this paper with some minor comments.

---

### Decision · Program_Chairs · 2024-06-17

Accept (Poster)